# Hierarchical Transformer for Electrocardiogram Diagnosis

**Xiaoya Tang**[1]                                                    XIAOYA.TANG@UTAH.EDU
**Jake Berquist**[1,2]                                             JAKE.BERGQUIST@UTAH.EDU
**Benjamin A. Steinberg**[3]                        BENJAMIN.STEINBERG@CUANSCHUTZ.EDU
**Tolga Tasdizen**[1]                                               TOLGA@SCI.UTAH.EDU

[1] *Scientific Computing and Imaging Institute, University of Utah, SLC, UT, USA*

[2] *Nora Eccles Harrison Cardiovascular Research and Training Institute, University of Utah, SLC, UT, USA*

[3] *University of Colorado Anschutz Medical Campus, Denver, CO, USA*

**Editors:** Accepted for publication at MIDL 2025

## Abstract

We propose a hierarchical Transformer for ECG analysis that combines depth-wise convolutions, multi-scale feature aggregation via a CLS token, and an attention-gated module to learn inter-lead relationships and enhance interpretability. The model is lightweight, flexible, and eliminates the need for complex attention or downsampling strategies.

**Keywords:** Multi-scale Transformer, ECG Classification, Depth-wise Convolution

## 1. Introduction

Cardiovascular diseases remain a significant health threat worldwide (Natarajan et al., 2020; Hu et al., 2022). Advances in computer-aided diagnosis have aimed to enhance the accuracy of electrocardiogram (ECG) interpretation and reduce associated costs (Dong et al., 2023). Existed deep learning methods applied to ECG tasks often involved elaborate convolutional and recursive structures (Yan et al., 2019). Given the sequential nature of ECG signals, the application of Transformers has shown promise due to their superior capacity to model long-range dependencies (Natarajan et al., 2020). Recent applications of Vision Transformers(ViT) in cardiac abnormality classification (Natarajan et al., 2020), arrhythmia detection (Hu et al., 2022) and phonocardiography-based valvular heart diseases detection (Jamil and Roy, 2023) have showcased their advantages in simplicity and training efficiency over CNNs and recurrent neural networks. Several recent studies have explored hierarchical transformers to address transformers' inductive bias limitations, by using shifted-window mechanisms (Li et al., 2021), employing transformer to bridge CNN encoders and decoders (Deng et al., 2021), integrating ResNet and channel attention (Wahid et al., 2024), and applying a deformable ViT (Dong et al., 2023) with depthwise separable convolutions. Although depth-wise convolutions and pyramid transformers have been noted, prior works haven't fully mined the inter-lead information nor optimized the use of hierarchical features. To address these gaps, we propose a novel hierarchical Transformer for ECG classification. Our main contributions are as follows:

- We use depth-wise convolutions to prevent the mixing of potentially important yet implicit information across leads before the transformer. This crucial aspect has been largely overlooked in previous works.

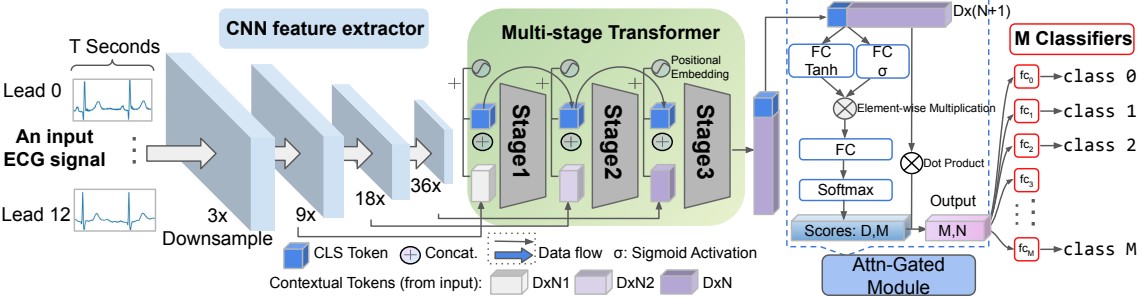

Figure 1: Framework: A Four-layer feature extractor with 1D depth-wise convolutions, three-stage transformer, and attention-gated module(from left to right). D is the embedding dimension, and N is the spatial size.

- We employ a CLS token to aggregate task-relevant information from multi-scale representations across stages. Instead of passing all output feature embeddings to the next stage, we propagate only the CLS token. By maintaining the same CLS token across the transformer, it aggregates multi-scale information.

- We integrate an attention-gated module to learn inter-leads associations. This module complements our model design alongside the depth-wise encoder.

## 2. Methodology

The framework is illustrated in Figure 1. Since the input ECG data is a multi-channel(lead) one-dimensional(1D) sequence, we employ a four-layer feature extractor consisting of 1D depth-wise convolutions (Howard, 2017). Each layer uses varying kernel sizes and strides to enable progressive downsampling, adapted from (Natarajan et al., 2020). To incorporate multi-scale inductive biases, we design a three-stage transformer, with each stage containing a stack of MSA layers. Each stage begins by integrating a feature map(contextual tokens in Figure 1) from the CNN with a learnable CLS token and ends by passing the CLS token to the next stage, where it is integrated with another feature map of reduced spatial size. By adjusting the input resolutions at each stage, we manipulate the receptive field from local to global. Information for each lead remains distinct and uncombined throughout the transformer. To model dependencies between leads, we apply an attention-gated module comprising three linear layers (highlighted in the blue dotted rectangle in Figure 1), inspired by (Chen et al., 2022). Please refer to Appendix A for further details. Code:https://github.com/xiaoyatang/3stageFormer.git.

## 3. Results and Analysis

We evaluate our model on two datasets: the 2020 PhysioNet/CinC Challenge dataset (Alday et al., 2020) and the KCL potassium classification dataset from Utah. The public dataset contains 43,101 recordings, and we follow a 10-fold validation for the multi-label classification of 24 diagnoses. As in the original challenge, we report macro $F_\beta$, $G_\beta$(reflecting precision and recall), and the challenge score which penalizes incorrect diagnoses. The KCL

dataset includes 54,419 training and 6,245 test recordings, with performance measured by macro-averaged AUC. Data processing and metric details are provided in Appendix B. Our models outperform previous PhysioNet/CinC Challenge winners(Prna) (Natarajan et al., 2020) across all metrics, rivals a two-phase method with contrastive pretraining, sCL-ST (Le et al., 2023), and confirm the efficacy and robustness of all variations of our model. Compared to 'Prna' which combines CNNs and transformers with a CLS token, our model's performance validates the multi-scale design and insightful use of the CLS token. We also incorporated a differential attention mechanism from language models (Ye et al., 2024), further enhancing performance by denoising attentions. Please refer to Appendix C and D for detailed analysis and interpretability insights with attention maps.

| Model | Fbeta | Gbeta | Challenge | Size(M) |
|---|---|---|---|---|
| LSTM | 0.4323 (±0.55%) | 0.2742 (±1.90%) | 0.4372 (±1.67%) | - |
| CNN | 0.4519 (±1.55%) | 0.2862 (±2.90%) | 0.4542 (±1.68%) | - |
| ResNet | 0.5088 (±0.41%) | 0.3278 (±2.69%) | 0.5158 (±0.80%) | - |
| ViT | 0.3263 (±1.65%) | 0.1970 (±1.88%) | 0.3197 (±2.44%) | - |
| Swin Transformer | 0.4812 (±0.87%) | 0.3045 (±0.66%) | 0.4811 (±1.41%) | - |
| BaT (Li et al., 2021) | 0.5011 (±0.68%) | 0.3125 (±1.15%) | 0.4958 (±0.83%) | - |
| Res-SE (Zhao et al., 2020) | 0.5607 (±1.30%) | 0.3264 (±2.94%) | 0.5939 (±0.30%) | 8.84 |
| SpatialTemporalNet | 0.4296 (±2.82%) | 0.2403 (±2.98%) | 0.4322 (±9.81%) | 4.52 |
| Prna (Natarajan et al., 2020) | 0.4975 (±5.17%) | 0.2679 (±6.99%) | 0.5463 (±3.23%) | 13.64 |
| Prna + CLS_Token | 0.5211 (±1.38%) | 0.2926 (±2.46%) | 0.5732 (±2.11%) | 13.64 |
| sCL-ST (Le et al., 2023) | 0.5100 (± − %) | **0.3622** (± − %) | 0.6053 (± − %) | - |
| Ours-No Attn_gated | 0.5672 (±0.60%) | 0.3296 (±3.34%) | 0.6174 (±1.06%) | 16.62 |
| Ours | 0.5778 (±0.76%) | 0.3407 (±2.18%) | 0.5980 (±0.85%) | 16.78 |
| Ours-No Attn_gated-Diff | 0.5704 (±1.19%) | 0.3397 (±0.50%) | **0.6203** (±1.23%) | 18.52 |
| Ours-Diff | **0.5850** (±1.15%) | 0.3459 (±0.38%) | 0.6063 (±0.18%) | 18.63 |

Table 1: Performances on multi-label ECG diagnoses. Results for the first six models are reported from (Li et al., 2021). Others were averaged over three folds in a 10-fold cross-validation following (Natarajan et al., 2020), unless otherwise indicated by their sources.

| Model | AUC | Size (M) |
|---|---|---|
| Prna (ViT) | 0.8126 (±1.08%) | 13.63 |
| Swin Transformer 1D-Tiny (Liu et al., 2021) | 0.7954 (±0.11%) | 47.47 |
| SpatialTemporalNet | 0.8218 (±0.35%) | 4.41 |
| Res-SENet | 0.8203 (±0.09%) | 8.82 |
| **Ours** | **0.8232** (±**0.33**%) | 10.94 |

Table 2: Performance comparison of typical models for KCL.

## 4. Conclusion

Our proposed hierarchical transformer efficiently manages diverse ECG tasks, enhances interpretability by leveraging multiscale features and innovative CLS token use, and flexibly adapts to different feature extractors, attention mechanisms, and input sizes.

## Acknowledgments

This work was supported by NIH R21HL172288. We thank Dr. Man Minh Ho for advice.

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

## Appendix A. Model Architecture

### A.1. Depthwise Convolutional Feature Encoder

Depthwise convolutions (Howard, 2017) employ a distinct filter for each input channel, capturing spatial relationships without cross-channel interactions. In the context of multi-lead ECG signals, these convolutions are applied individually to each lead. Subsequently, the resulting feature maps from each lead can be transformed separately onto a new space (Dong et al., 2023).

### A.2. Three-Stage Transformer

According to previous researches, the effective receptive field of ViT shifts from local to global as it progresses through the layers. We structure a transformer encoder into three stages, each containing a stack of MSA layers, with the division of layers tailored to specific needs. Our approach involves feeding hierarchical feature embeddings(called contextual tokens in 1) into three stages, derived from different layers of our convolutional encoder using three distinct downsampling rates from the input ECG segment. After each stage, the CLS token is extracted and concatenated with a new sequence of embeddings at a larger downsampling rate, then passed into the next transformer stage. This progressive feeding of downsampled features compels the model to transition its focus from detailed to more abstract, global patterns. Utilizing the CLS token allows us to efficiently aggregate and transfer multi-scale information to the final classification layer.

### A.3. Attention-Gated Module

Given an output from three-stage transformer $x$ with dimensions $x \in \mathbb{R}^{B \times C \times S}$, where $B$ represents the batch size, $C$ the number of channels, and $S$ the sequence length. The information for each lead remains distinct and uncombined. Thus we utilize an attention-gated module to model dependencies between leads, inspired by (Chen et al., 2022). This module comprises three linear layers designed to uncover latent dependencies between channels, which correspond to associations between ECG leads in this context. The attention score $a$

is computed through an element-wise multiplication of the query and key vectors, resulting in $a \in \mathbb{R}^{B \times C \times S}$, as shown in Eq. 1. $W_q \in \mathbb{R}^{S \times S}$, $b_q \in \mathbb{R}^S$, $W_k \in \mathbb{R}^{S \times S}$, and $b_k \in \mathbb{R}^S$ represent the weights and biases of the linear layer for learning query and key, with $\sigma$ denotes the Sigmoid function.

$$
\begin{aligned}
q &= \tanh(W_q x + b_q) \\
k &= \sigma(W_k x + b_k) \\
a &= q \odot k
\end{aligned}
\tag{1}
$$

A linear project is applied to the attention scores resulting in $a' \in \mathbb{R}^{B \times C \times N}$, where $N$ is the number of classes. The raw attentions are then normalized by a softmax and multiplied with the output from the three-stage transformer, yielding $v \in \mathbb{R}^{B \times N \times S}$, shown in Eq.2. These operations are analogous to the MSA mechanism. Finally, a separate classifier for each class is applied across the sequences, where $v_i$ denotes the segment of $v$ corresponding to the $i$-th class.

$$
\begin{aligned}
a' &= \text{Projection}(a) \\
a'' &= \text{softmax}(\text{transpose}(a', (0, 2, 1))) \\
v &= a'' @ x \\
\text{logits}_i &= W_i v_i + b_i \quad \text{for each } i \in \{1, \ldots, N\}
\end{aligned}
\tag{2}
$$

## Appendix B. Data and Evaluation Metrics

We utilize the public training data from the 2020 PhysioNet/CinC Challenge (Alday et al., 2020) and KCL data from our group. The public dataset comprises $43,101$ recordings, and we adopt the 10-fold split used by the winner model 'Prna' (Natarajan et al., 2020). This setup involves a multi-label classification task related to 24 diagnoses. Following the preprocessing steps of 'Prna', we resample all recordings to $500Hz$, apply an FIR bandpass filter, and perform normalization. We also randomly crop multiple fixed-length ECG segments of $T = 15$ seconds from the input, adding padding when necessary for segments shorter than 15s. We also leveraged the wide features that they used. For evaluation metrics, we report macro $F_\beta$, $G_\beta$, geometric mean(GM) combining precision and recall and the challenge score defined by the challenge organizers (Alday et al., 2020), detailed in Eq. 3. The score $S$ generalizes standard accuracy by fully crediting correct diagnoses and penalizing incorrect ones based on the similarity between arrhythmia types. Here $a_{ij}$ represents an entry in the confusion matrix corresponding to the number of recordings classified as class $c_i$ but actually belonging to class $c_j$, with different weights $w_{ij}$ assigned based on the similarity of classes $c_i, c_j$:

$$
\begin{aligned}
F_\beta &= (1 + \beta^2) \cdot \frac{TP}{(1 + \beta^2) \cdot TP + FP + \beta^2 FN} \\
G_\beta &= \frac{TP}{TP + FP + \beta FN}, \beta = 2 \\
S &= \sum_{ij} w_{ij} a_{ij}
\end{aligned}
\tag{3}
$$

For the KCL potassium classification, all recordings maintain a uniform sampling rate of $500Hz$. After applying normalization, we randomly crop these to fixed segments of $T = 5s$. The dataset includes $54,419$ recordings for training and $6,245$ for testing. We report the macro-averaged area under the receiver operating characteristic curve (AUC) on test data.

## Appendix C. Result Analysis

Our model surpasses previous winners in the 2020 PhysioNet/CinC Challenge dataset, Prna (Natarajan et al., 2020) and Res-SENet (Zhao et al., 2020), and other commonly used architectures. Results are shown in 1. An important observation from these results is the enhanced performance of the standard ViT model, Prna, upon integration of a CLS token, which validates the CLS token's significance in classification tasks. By adjusting the input resolutions at each transformer stage, we manipulate the receptive field of attention—from local, with larger spatial-sized embeddings, to global, with smaller spatial-sized embeddings. These variations in granularity are achieved by modifying the kernel sizes and strides in the CNN feature extractor. We hypothesize that passing only the CLS token between stages, rather than all output embeddings, focuses the model on prediction-relevant information. Through backpropagation, the CLS token can access crucial details from the key and value representations in the attention mechanism. By maintaining the same CLS token across the transformer, it aggregates multi-scale information, thus enhancing the model's inductive bias. By outperforming the integration of Prna with a CLS token, we confirm our model's effectiveness. Additionally, our model demonstrates that both with and without the Attention_gated module, it maintains competitive performance, demonstrating the efficiency of our three-stage transformer.

To further validate the efficiency and generalizability of our approach, we conducted additional tests on the KCL binary classification task, comparing our model against other prominent architectures. Our model showcased the highest AUC, outperforming models such as SpatialTemporalNet and ViT (Prna), shown in 2. These results confirm the robustness and adaptability of our model, effectively identifying complex patterns essential for precise ECG classification.

### C.1. Differential Attention Mechanism

Inspired by attention denoising techniques in large language models and the similar sequential nature of ECG signals and language data, we adapted the differential attention mechanism from (Ye et al., 2024). This mechanism enhances focus on relevant contexts while suppressing noise by subtracting two softmax-transformed attention maps, which reduces noise and promotes the development of distinct attention patterns, as delineated in Eq.4. Given an input $X \in \mathbb{R}^{N \times d}$, we project it to query, key, and value tensors $\mathbf{Q}_1, \mathbf{Q}_2, \mathbf{K}_1, \mathbf{K}_2, \mathbf{V} \in \mathbb{R}^{N \times 2D}$. The differential attention operation DiffAttn$(\cdot)$ then computes the outputs. A significant adaptation for ECG analysis involves omitting the upper diagonal mask, originally used in language generative models to prevent attending to future tokens, thus allowing the model

to consider the entire ECG sequence simultaneously.

$$[\mathbf{Q}_1; \mathbf{Q}_2] = XW^Q, \quad [\mathbf{K}_1; \mathbf{K}_2] = XW^K, \quad \mathbf{V} = XW^V$$

$$\text{DiffAttn}(X) = \left( \text{softmax} \left( \frac{\mathbf{Q}_1 \mathbf{K}_1^T}{\sqrt{D}} \right) - \lambda \text{softmax} \left( \frac{\mathbf{Q}_2 \mathbf{K}_2^T}{\sqrt{D}} \right) \right) \mathbf{V} \tag{4}$$

## Appendix D. Interpretability

The multi-head self-attention(MSA) allows each head to learn distinct attention patterns across the time sequence. These patterns can be analogized to distinct attentions across different ECG leads, facilitated by our depthwise encoder. During the evaluation phase of the KCL classification, we randomly selected an abnormal sample, with the attention map at the final stage shown in 2 and 3. We qualitatively assessed the attention map by examining which areas of the ECG signals garnered the highest attention in this unhealthy case. Notably, our model exhibited heightened attention to clinically significant features such as the QRS complex, S-T segment, and T-wave, which are recognized as clinical indicators of changes in serum potassium levels. The proposed approach also underscores how the model's attention shifts across different stages. While we do not leverage lead attention here, the attn_gated module after MSA allowed us to discern dependencies among multiple leads. This capability further provides valuable insights into how the model relies on different leads, enhancing our understanding of deep learning models for ECG diagnosis.

Figure 2: Attentions in an abnormal case (high potassium) for leads 1 to 4, illustrating final stage attentions.

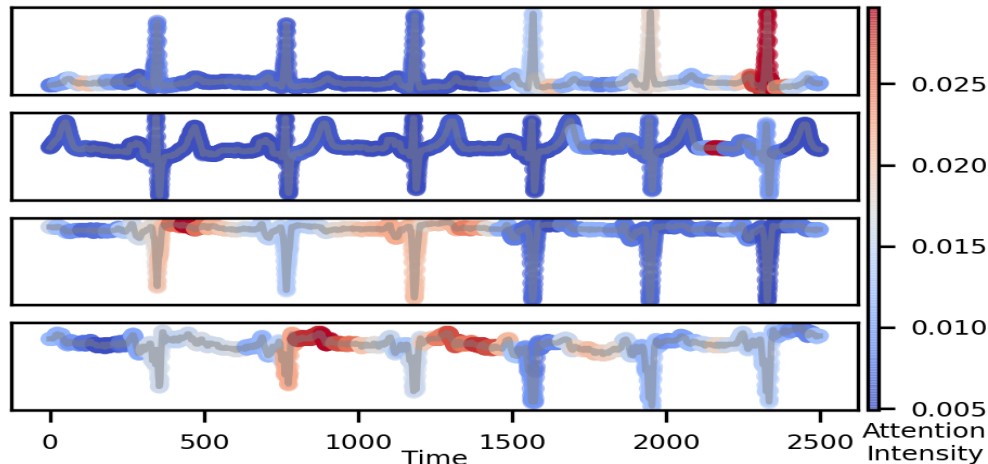

Figure 3: Attentions in an abnormal case (high potassium) for leads 5 to 8, illustrating final stage attentions.

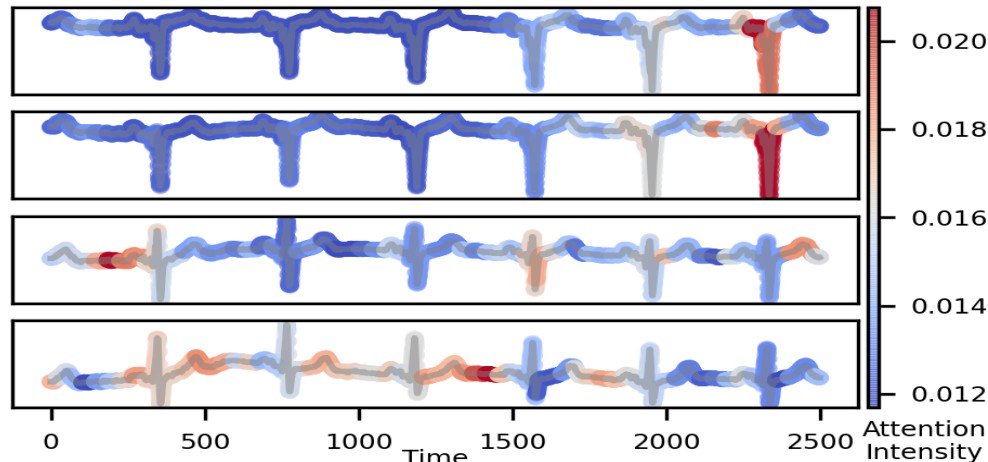

