# OpenReview forum: "Hierarchical Transformer for Electrocardiogram Diagnosis"
_MIDL.io/2025/Short_Papers — MIDL 2025 - Short Papers_

### Official Review · Reviewer_VxR7 · 2025-04-24

**Rating:** 4
**Confidence:** 5

**Summary:**

interesting paper combining prior concepts for cardiovascular disease classification from ECG data. I believe for a short paper it has some merit and would generate important discussions during the meeting.

**Strengths:**

Evaluation against different baselines
Source code is made available
Propagating only the CLS token between stages is interesting

**Weaknesses:**

its a larger size model compared to some other baselines that have similar accuracy

---

### Decision · Program_Chairs · 2025-05-01

Accept